# Efficient Asset Allocation: Application of Game Theory-Based Model for Superior Performance

Mirza Sikalo *, Almira Arnaut-Berilo and Azra Zaimovic

School of Economics and Business, University of Sarajevo, Trg Oslobodjenja—Alija Izetbegovic 1, 71000 Sarajevo, Bosnia and Herzegovina; almira.arnaut@efsa.unsa.ba (A.A.-B.); azra.zaimovic@efsa.unsa.ba (A.Z.)
* Correspondence: mirza.sikalo@efsa.unsa.ba

**Abstract:** In this paper, we compared the models for selecting the optimal portfolio based on different risk measures to identify the periods in which some of the risk measures dominated over others. For decades, the best known return-risk model has been Markowitz's mean-variance model. Based on the criticism of the classical Markowitz model, a whole series of risk measures and models for selecting the optimal portfolio have been developed, which are divided into two groups: symmetrical and downside risk measures. Based on the tools provided by game theory, we presented a minimax model for selecting the optimal portfolio based on the maximum loss as a measure of risk. Recent research has shown the adequacy of the application of this risk measure and its dominance concerning variance in certain circumstances. Theoretically, the model based on maximum loss as a measure of risk relies on a much smaller number of assumptions that must be satisfied. In the empirical part of the paper, we analyzed the real return performance, structure, correlation, stability, and predictive efficiency of the model based on maximum loss return as a measure of risk and compared it with the other famous models to determine whether the maximum loss-based risk measure model is more suitable for use in certain circumstances than conventional return-risk models. We compared portfolios created based on different models over the period of 2000–2020 from a selected sample of stocks that are components of the STOXX Europe 600 index, which covers 90% of the free market capitalization in the European capital market. The observed period included 3 bear market periods, including the period of market decline during the COVID-19 crisis. Our analysis showed that there was no significant difference in portfolio returns depending on the selected model using the "buy-and-hold" strategy, but there were crisis periods. The results showed a significantly higher stability of portfolios selected on the criterion of minimizing the maximum loss than others. In periods of market decline, this portfolio achieved the best performance and had a shorter recovery period than others. This allowed superior use of the minimax model at least for investors with a pronounced risk aversion.

**Keywords:** portfolio risk; portfolio diversification; portfolio optimization; game theory; COVID-19 crisis

## 1. Introduction

The problem of choosing the optimal portfolio has been one of the basic issues in finance for decades, which is based on the trade-off between risk and return. Until the middle of the last century, however, people were not able to quantify risk. They believed that it was only necessary to diversify the portfolio sufficiently and that, in this way, risk would be minimized. This approach is known as traditional portfolio theory (Nawrocki 1999). Markowitz's (1952) work not only quantified risk but also made it possible to measure the impact of the risk of each asset on the risk of the entire portfolio, which is known as modern portfolio theory. Markowitz's initial theory is based on a number of simplifying assumptions and conditions that are extremely difficult to satisfy in practice. Despite its widespread use in practice, the modern portfolio theory and Markowitz's optimization

approach, which is based on quadratic programming and the first two moments of the distribution of return probability as the main parameters, have been met with criticism.

As a result, in recent decades, the standard mean-variance approach has been modified by applying more appropriate risk measures in the optimization algorithm, and many models for selecting the optimal portfolio and measuring portfolio performance have been developed, aiming at simplifying the portfolio selection problem and making it applicable in real investment practice. In this way, a whole set of theories and models were created, which Sharma et al. (2017) called "giant tree whose seed was farmed by Markowitz." All of these approaches are commonly referred to as post-modern theory and all of these models are primarily determined by the parameter used in them as a risk measure. Immediately after they were introduced, a natural question was asked: Which of these risk measures is the best?

In this paper, we conducted a study comparing models based on different risk measures: symmetric, such as variance, and lower risk measures, such as minimum return.

The paper is divided into five sections. Section 2 provides an overview of the literature in which we review a set of most commonly used measures for selecting the optimal portfolio. This section highlights the most significant features of each of these models, as well as the advantages and disadvantages in theory and practice. In Section 3, we explain the methodology and data used for the analysis. Section 4 presents the results with the performance of the models in the verification period and the verification of their predictive efficiency with the final consideration. Finally, concluding remarks and further directions for research are given in Section 5.

## 2. Theoretical Background with Literature Review

Markowitz (1952) presented a mathematical framework for portfolio selection known as the mean-variance model, because it is based on average return as a measure of expected return and variance, or standard deviation, as a measure of risk. The basis of Markowitz's theory is that for a given level of risk, a rational investor chooses assets that realize a higher return, or, for a given level of return, he chooses assets whose returns are less risky. Assets and portfolios that satisfy this form an efficient frontier and all others are inefficient. In addition, Markowitz highlights the impact of individual securities on the risk of the entire portfolio and quantifies this impact using covariance. Markowitz's optimization model is a problem of quadratic programming whose solutions are the weights $w_i$, $i = \overline{1, N}$. The goal is to minimize the risk of the portfolio expressed by the variance with a predetermined expected rate of return C, the conditions of the budget constraint, and the impossibility of short selling (mathematical constraint of nonnegativity).

$$\min \sigma_P^2 = \sum_{j=1}^{N} \sum_{i=1}^{N} w_i w_j Cov_{i,j} \tag{1}$$

$$\sum_{i=1}^{N} w_i \overline{R_i} = C \tag{2}$$

$$\sum_{i=1}^{N} w_i = 1 \tag{3}$$

$$w_i \geq 0, \ i = \overline{1, N} \tag{4}$$

The fundamental problem of applying any model to select the optimal portfolio is caused by the inability to accurately predict all possible future returns. The use of historical returns, which are the most common approximation of possible returns, is irrelevant for predicting the future (Ibrahim et al. 2008). Given that Markowitz's model crucially depends on the quality and accuracy of the input data and a small change in the value of some of them may affect the outcome, it is questionable to what extent variance is a stable measure of risk (Chow et al. 1999). Given the possible errors in estimating input parameters, the

Markowitz model is often viewed as a good model for obtaining an optimal solution, but with erroneous data (Michaud 1989). Yuan et al. (2015) found that even small deviations in the future from historical returns violate the assumption of exact knowledge of expected returns, variances for stocks, as well as covariances between them.

Adequacy of standard deviation as a risk measure is also questionable, as investors do not view positive deviations from expected returns as a bad thing. Therefore, the definition of risk as a deviation of actual from expected return was very quickly corrected to include only the negative deviation of the actual from the expected return, which is called semivariance (Strong 2006). However, the condition of symmetry of the distribution equals the probability of positive and negative deviations, so this adjustment has no real effect and, mathematically, the standard deviation is never inferior to the semivariance. As long as the assumption of normality of return, as one of the basic assumptions of the MV model, is fulfilled, this condition is satisfied. Otherwise, the MV model can provide poor and illogical solutions. However, the assumption of return normality, even if historical returns correctly depict the future, is very rarely fulfilled (Young 1998).

Another problem with the MV model is the fact that a longer period of time needs to be observed to better evaluate the parameters, which has two major negative consequences for the efficiency of the model. First, over longer periods, due to compound returns, a normal distribution is not an accurate estimate of the distribution of securities' return probabilities. Instead, a lognormal distribution is more appropriate, which assigns higher probability to extremely high values compared to extremely low values of a random variable (Levy 1973). This distorts the symmetry of the distribution. On the other hand, observing a longer time period also implies a greater possibility that in the future, the overall market movement will change. In contrast, the assumption of the MV model is stationarity of the time series, i.e., invariance of variance and expected returns over time. Blattberg and Gonedes (1974), Castanias (1979), and Christie (1982) have also shown that the variance of returns changes over time, i.e., that the assumption of homoskedasticity of returns is violated. In some periods, the reactions and changes in return are greater than in others. The fact that return variability changes over time was first established by Mandelbrot (1963), who observed that the tails of the return distribution are longer than normal.

From the introduction of Markowitz's model until today, many authors in finance have focused on developing new models, which included many investor requirements in terms of return, risk, portfolio structure, investment time, transaction costs, and the like. One of the main goals of all previous research in modern portfolio theory was the setting of alternative risk measures (Anagnostopoulos and Mamanis 2010) and, as Markowitz's initial model is a quadratic programming problem, many have aspired to linearize the problem of portfolio optimization in this way. In this sense, some of the measures are based on the same principle as the MV model, where risk is considered as a deviation of the actual return from the expected return. The most famous among them is the MAD model (Konno and Yamazaki 1991). In this model, variance has been replaced by mean absolute deviation as a measure of risk, which has made the optimal portfolio selection model a problem of linear programming. On the other hand, risk can be viewed as the possibility of not achieving a certain return, which is called the shortfall risk measure (Mansini et al. 2014). Nawrocki (1999) pointed out that downside risk measures are supposedly a major improvement over traditional portfolio theory. These measures quantify risk from different perspectives. The two most significant models representing downside risk measures are minimax (Young 1998) and CVaR (Rockafellar and Uryasev 2000). Minimax is an alternative method of choosing the optimal portfolio based on game theory. The selection of the optimal portfolio can be represented as a noncooperative game with two players. Based on historical returns, it is possible to form a payoff matrix with an investor as a first player and the market as a second player. The proportions invested in the shares of different issuers are possible strategies for the investor, while the repetition of the past represents strategies

for the market. The linear programming model derived from these assumptions can be represented as follows:

$$\max R \tag{5}$$

$$\sum_{i=1}^{N} R_{ij} \cdot w_i \geq R, \, j = \overline{1, m} \tag{6}$$

$$\sum_{i=1}^{N} w_i = 1 \tag{7}$$

$$w_i \geq 0, \, i = \overline{1, N} \tag{8}$$

Young (1998) developed their minimax model on a similar basis, adding a constraint of the lowest acceptable return, and explaining the possibility of including transaction costs and taxes in the model. With the minimax model, there is no assumption of a normal return distribution. We conclude that the minimax model has an advantage when returns are not normally distributed. However, under conditions where returns follow a normal distribution, these two risk measures will give approximately equal results. Consequently, for any portfolio with a given average return, the portfolio with the lowest variance simultaneously represents the portfolio with the highest minimum return. The reason the MV model works outside the principle of rationality, in this case, is that the investor who has risk aversion through the portfolio wants to avoid low returns, while MV gives equal importance to a deviation of the actual from the expected return, even if the returns are high enough. Therefore, the minimax rule has an advantage over the MV model, as it gives good results regardless of the distribution (Young 1998).

On the other hand, Morgan JP presented Value-at-Risk (VaR) in 1994 as a synthetic measure of risk that measures the exposure of a particular asset or portfolio to the possibility of loss. However, VaR had a major drawback—this measure did not observe the principle of subadditivity, so it was not coherent, meaning that the effects of diversification on risk do not exist (Artzner et al. 1999). As a consequence, the Conditional Value-at-Risk (CVaR) measure was introduced by Rockafellar and Uryasev (2000), which can be used by a linear risk optimization model. CVaR measures the expected value of losses that exceed the value of VaR and attempts to minimize it.

With the exception of the MV model, none of these models pay attention to the correlation coefficients, which makes computation easier. This is supported by the fact that in times of financial crisis, correlation coefficients tend to 1 and the benefits of diversification disappear (Campbell et al. 2002). A similar behavior of correlation coefficients is predicted by Konno and Yamazaki (1991). They find that in cases where the overall market is declining, they state that correlations between stocks become irrelevant. In this case, these models have a theoretical advantage over the MV model. Moreover, in times of crisis, investors are generally more afraid of losses, so it is more appropriate to consider risk in this way.

Kalayci et al. (2019) provided a comprehensive review of deterministic models that included 175 papers. They cited the work Konno and Yamazaki (1991), Young (1998), and Krokhmal et al. (2001) as the most important papers that point out the shortcomings of mean-variance models. As one of the conclusions, they noted that most researchers have tried to present an algorithm for decision making and that it is necessary to find ways to measure performance differently to verify portfolio efficiency in practice.

Righi and Borenstein (2018) compared 11 measures from different classes in the U.S. capital market. For each time period, they ranked portfolios according to performance from best to worst and used nonparametric tests to test for differences. They showed that there is no dominant risk measure.

Hunjra et al. (2020) considered mean-variance, semivariance, mean absolute deviation, and CVaR as risk measures comparing their performance to understand their appropriateness for effective portfolio management for investors using return data for stocks listed in Pakistan, India, and Bangladesh. They analyzed the structure and realized returns

and showed that the downside risk measure CVaR gives the best results in all scenarios, supporting the conclusion that variance is not an appropriate risk measure for all markets and all economic scenarios.

Subramaniam and Chakraborty (2021) analyzed the impact of investors' mood related to the COVID-19 pandemic on stock market returns by creating a unique COVID-19 fear index. The study highlighted that COVID-19 fear strongly impacts the stock market and found a strong negative association between COVID-19 fear and stock returns. Increased fear and volatility are caused by the fact that the media plays a role in transmitting financial contagion during the COVID-19 pandemic, which was especially pronounced at the beginning of the pandemic (Akhtaruzzaman et al. 2022). On the same track, Shehzad et al. (2020) showed that there are differences in volatility depending on the market in this period. Similarly, Duttilo et al. (2021) investigated the impact of two COVID-19 waves on return volatility in the Euro area stock market. The results revealed that Euro area stock markets responded differently to the COVID-19 pandemic. Specifically, the first wave of COVID-19 infections had a notable impact on stock market volatility of Euro area countries with middle-large financial centers, while the second wave had a significant impact only on the stock market volatility of Belgium. However, spillovers were transferred between countries during the COVID-19 period, which increased the hedging costs to optimize portfolios (Akhtaruzzaman et al. 2021).

Delis et al. (2021) examined the impact of the coronavirus crisis on returns and risk. They assumed that the coronavirus market collapse was much deeper than the previous crisis, but that due to the exogenous nature of the crisis, the recovery was much faster. The results of their study showed that the coronavirus crisis caused a strong negative reaction on the skewness and total market price of risk, which was even more negative than other financial crises in history. On this basis, they pointed to the need to adjust variance as a measure of risk.

### 3. Materials and Methods

The main purpose of the research was to identify potential differences between the models and to determine the conditions under which targeting downside risk instead of variance actually works in practical applications by testing the following hypotheses:

**Hypothesis 1 (H1).** *There are significant differences in portfolio structure depending on which portfolio selection model is chosen based on different risk measures.*

**Hypothesis 2 (H2).** *Models based on lower risk measures have better predictive power than models based on symmetric risk measures, especially in crisis and post-crisis periods.*

The portfolio models used in the study were: (1) MV1—minimum variance portfolio model, (2) MV2—the portfolio model with the highest Sharpe ratio, (3) Minimax model based on game theory, (4) CVaR model, and (5) MAD model. MV1, MV2, and MAD represent symmetric risk measures, and minimax and CVaR represent downside risk measures. All models are listed in Table 1.

**Table 1.** List of various asset allocation models.

| Objective Function | Constraints | Symbols Description |
|---|---|---|
| | **1**    Minimum variance (MV1) | |
| $\min\left(\sum\limits_{i=1}^{N}\sum\limits_{j=1}^{N} w_i w_j Cov_{i,j}\right)$ | $\sum\limits_{i=1}^{N} w_i = 1,\ w_i \geq 0,$ | N—number of shares <br> $w_i$—weights in shares i <br> $Cov_{i,j}$—covariance of returns between shares i and j |
| | **2**    Maximum Sharpe ratio (MV2) | |
| $\max\left(\frac{R_p - r_f}{\sigma_p}\right)$ | $R_p = \sum\limits_{i=1}^{N} w_i \overline{R}_i$ <br> $\sigma_p = \sqrt{\sum\limits_{i=1}^{N}\sum\limits_{j=1}^{N} w_i w_j Cov_{i,j}}$ <br> $\sum\limits_{i=1}^{N} w_i = 1,$ <br> $w_i \geq 0,\ i = \overline{1, N}$ | $R_p$—expected (mean) portfolio return <br> $r_f$—risk-free rate <br> $\sigma_p$—portfolio standard deviations <br> $\overline{R}_i$—expected return on shares *i* <br> N—number of shares |
| | **3**    Minimax model | |
| maxR | $\sum\limits_{i=1}^{N} R_{it}\cdot w_i \geq R,\ t = \overline{1, m}$ <br> $\sum\limits_{i=1}^{N} w_i = 1$ <br> $w_i \geq 0,\ i = \overline{1, N}$ | $R_{it}$—returns on shares i in period t <br> m—number of periods <br> N—number of shares <br> R—portfolio return that is calculated as the maximum lower boundary for all possible investment weight combinations |
| | **4**    CVaR model | |
| $\min\frac{1}{m\alpha}\sum\limits_{t=1}^{m} d_t - VaR$ | $d_t = \max\left(VaR - \sum\limits_{i=1}^{N} R_{it}\cdot w_i,\ 0\right)\ t = \overline{1, m}$ <br> $\sum\limits_{i=1}^{N} w_i = 1$ <br> $w_i \geq 0,\ i = \overline{1, N}$ | VaR —Value at risk <br> $\alpha$—the probability of a return less than VaR is (at least) $\alpha$ (we used $\alpha = 5\%$) <br> $d_t$—deviations below the VaR |
| | **5**    MAD model | |
| min MAD | $MAD = \sum\limits_{i=1}^{N} w_i |R_i - \overline{R}_i|$ <br> $\left|\sum\limits_{i=1}^{N} (R_{it} - \overline{R}_i)\cdot w_i\right| \leq MAD,\ t = \overline{1, m}$ <br> $\sum\limits_{i=1}^{N} w_i = 1,$ <br> $w_i \geq 0,\ i = \overline{1, N}$ | $w_i$—weights in shares i <br> $R_{it}$—returns on shares i in period t <br> $\overline{R}_i$—expected return on shares i <br> m—number of periods <br> N—number of shares |

For this research, a sample of 10% of the stocks from the population was selected using the method of simple random sampling based on the components that make up the STOXX Europe 600 Index at the end of 2020. The STOXX Europe 600 is an index composed of 600 components representing small, medium, and large companies from 17 European countries and covers 90% of the free market capitalization in the European region capital market, so we can assume that this index adequately approximates the movement of the entire European market. Due to missing data and nonsynchronous trading issues or the fact that some stocks were not publicly traded during the entire observed period, the final number of stocks in the sample was 57. The stocks included in the sample are listed in Table 2.

**Table 2.** List of stocks.

| Stock Exchange | Stocks |
| --- | --- |
| London SE | FERG.L, SGRO.L, AVV.L, RR.L, BKG.L, RMV.L, ITV.L, SXS.L, GSK.L, IMB.L, SVT.L, AAL.L, RIO.L |
| DB Xetra | BAS.DE, FRE.DE, HNR1.DE, HEI.DE, VNA.DE, CBK.DE, RHM.DE, DAI.DE, RWE.DE |
| Euronext Paris | MC.PA, SAN.PA, BN.PA, RNO.PA, ATO.PA, ICAD.PA, AC.PA, EL.PA |
| Stockholm SE | VOLV-B.ST, NIBE-B.ST, TEL2-A.ST, SKA-B.ST, ERIC-A.ST |
| Euronext Amsterdam | AKZA.AS, AGN.AS, HEIA.AS, RDSA. AS, ING.AS |
| SIX Swiss Exchange | NOVN.SW, SREN.SW, CLN.SW, BEAN.SW, GIVN.SW, NESN.SW |
| Madrid SE | BBVA.MC, ANA.MC, REP.MC |
| Italian Bourse | EXO.MI, ISP.MI, ENI.MI |
| Copenhagen SE | NOVO-B.CO, DANSKE.CO |
| Euronext Brussels | SOF.BR |
| Vienna SE | OMV.VI |
| Oslo SE | SALM.OL |

Based on historical weekly prices, we calculated discrete returns over the period from 2000 to 2019 and divided this period into 20 parts—each year represents one period. Based on these data, we created 20 sets of 6 portfolios. We tested the success, stability, and predictive power of the model in the following period (from 2001 to 2020, for an appropriate model).

For portfolio weights in the different model calculations, we used our own model-based algorithm implemented in Visual Basic for Applications and run in an Excel environment, and for model assumptions, comparisons, and conclusions, we used statistical software. Testing the hypothesis that stock returns follow a normal distribution, descriptive statistics, and correlation analyses were performed in STATA, and appropriate nonparametric tests for comparing performance were performed in SPSS.

Following the above, we compared the deviations in the theoretical predictions given by the MV and minimax models with the realized values. Cheng and Wolverton (2001) pointed out problems in comparing portfolios that use different risk measures as criteria. Each model will give the best results in its own space and any other portfolio will be inferior. Therefore, we conducted a five-part analysis. The structure of the portfolio was first compared by examining whether there is a difference in the number of shares according to different approaches. We then performed a portfolio return analysis by comparing the value of the portfolio formed at the beginning of the year throughout the year for all 20 periods observed. We extended this analysis with a correlation analysis with special reference to the years of market decline. In the last two sections, we examined the stability and predictive power of each of the models. We used the Sharpe ratio (Sharpe 1994) to test for stability. Although many other measures for portfolio stability have been developed over the years (Sharma et al. 2017), the Sharpe ratio is still the most widely used and best understood. To measure predictive power, we compared the returns again. For the minimax model, the minimum return is also a risk, so we compared the lowest realized return with this value, but for other portfolios, we can calculate the actual return and compare it with the expected one.

During the analysis period, the market went through several phases (Figure 1). From the beginning of the observed period in 2000 until the first quarter of 2003, the value of the Euro Stoxx 600 Index showed a negative trend. During this period, which spanned 150 weeks, the index lost 60% of its value, which is known as the early 2000s recession. In the period that followed, the market recovered and grew, reaching the same value in mid-2007 as in the early 2000s. This was followed by a period of global financial crisis, and the index lost 60% of its value again in less than 90 weeks. The recovery, with minor

periods of instability, lasted 300 weeks. This was followed by a period of stability with minor fluctuations until the first quarter of 2020 when the index lost a third of its value in 4 weeks due to the COVID-19 pandemic. A special focus of the work was on the periods of market decline, i.e., bear markets in 2002, 2007–2008, and 2020.

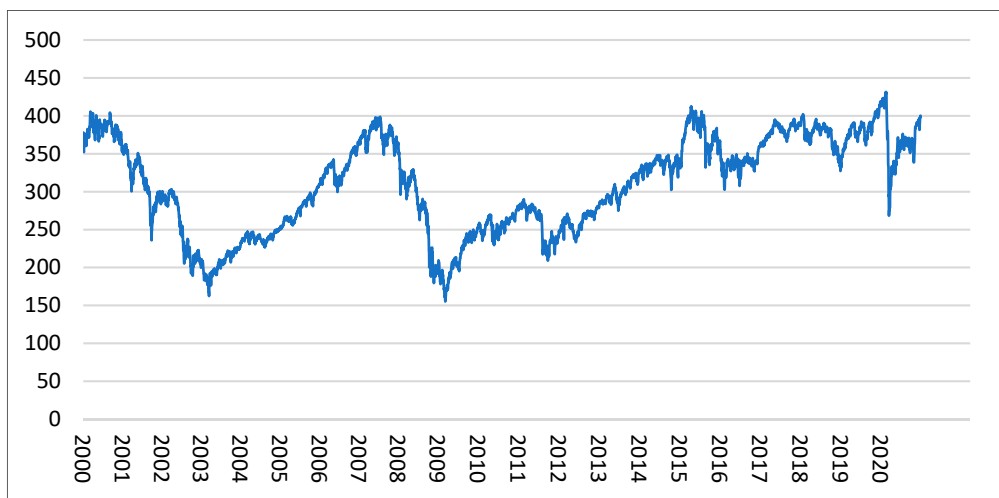

**Figure 1.** Change in the value of STOXX Europe 600 index during the observed period.

## 4. Results and Discussion

As we have already mentioned, the basic problem in the practical application of the MV model relates to the fact that the presence of higher moments of distribution significantly affects the formation of the portfolio. Under conditions when there are higher moments, the variance is not a representative measure of risk. Based on the conducted Jarque–Bera test, we conclude that only several stocks from the sample meet the assumption of normal distribution of returns in the observed periods and that most stocks have pronounced negative asymmetry, which indicates the appropriateness of using alternative risk measures.

### 4.1. Portfolio Structure Analysis

Table 3 shows the average number of stocks in the portfolio structure based on different models and criteria over a twenty-year period. We can see that both portfolios based on the MV model in 11 of the 20 periods have more stocks in their structure compared to all other models and, in 19 of the 20 periods, at least one MV model has more stocks in its structure than any other. We tested the hypothesis that there are differences in the number of stocks depending on the chosen optimization model. The Kruskal–Wallis test performed confirmed the hypothesis that the average number of shares in the selected portfolios is different (Appendix A). As it is based on a different principle, the portfolio structure chosen by the minimax model is the most diverse compared to the others. This is consistent with the findings of research by Byrne and Lee (2004) and will lead to different results when measuring the portfolio performance achieved. In addition, Keyzer and Schaepmeester (2014) suggested that individual investors need a smaller number of stocks for a well-diversified portfolio with higher correlations to the market. This conclusion could be even more significant if a multi-period model with transaction costs is considered as an additional drawback of the mean-variance model.

Zaimovic et al. (2021) found that the number of stocks leading to optimal diversification is impacted by a variety of factors: the way risk is measured, size, asset classes and features of the asset classes, investor characteristics, the model of diversification (i.e., equally weighted or optimal allocation), frequency of data, time horizon, market conditions, etc. Our results are consistent with their findings. The number of stocks required to diversify risk differs depending on the optimization model and risk measure chosen.

**Table 3.** Number of stocks in the structure of selected portfolios.

| Year 20– | 00 | 01 | 02 | 03 | 04 | 05 | 06 | 07 | 08 | 09 | 10 | 11 | 12 | 13 | 14 | 15 | 16 | 17 | 18 | 19 | Average |
|---|---|---|---|---|---|---|---|---|---|---|---|---|---|---|---|---|---|---|---|---|---|
| M1 | 15 | 9 | 14 | 13 | 25 | 9 | 24 | 12 | 14 | 23 | 13 | 30 | 19 | 18 | 17 | 9 | 7 | 14 | 12 | 16 | 15.90 |
| M2 | 22 | 6 | 12 | 10 | 14 | 20 | 20 | 17 | 32 | 21 | 14 | 7 | 10 | 19 | 16 | 25 | 6 | 8 | 10 | 5 | 15.30 |
| Minmax | 11 | 14 | 16 | 9 | 12 | 12 | 12 | 11 | 11 | 13 | 6 | 9 | 9 | 7 | 12 | 4 | 7 | 10 | 11 | 10 | 10.30 |
| CVaR | 11 | 18 | 18 | 21 | 17 | 10 | 9 | 9 | 11 | 7 | 5 | 8 | 9 | 6 | 12 | 9 | 8 | 16 | 12 | 13 | 11.45 |
| MAD | 11 | 7 | 9 | 13 | 11 | 9 | 9 | 11 | 9 | 14 | 7 | 11 | 6 | 6 | 2 | 6 | 8 | 16 | 18 | 10 | 9.65 |

Notation: We excluded stocks with a portfolio weight of less than 0.01% from the analysis.

*4.2. Portfolio Return Analysis*

We continued the analysis by comparing the returns achieved by the portfolio selected using different models, given that the primary interest of each investor is to achieve the highest possible return. We calculated the average return of the created portfolio depending on the holding period, i.e., the assumption is that we buy the portfolio at the beginning of the observed period and hold it for a year. No portfolio has consistently better performance than other portfolios (Table 4).

**Table 4.** Results of Kruskal–Wallis one-way ANOVA on ranks test for average return by portfolio.

| Independent-Samples Kruskal–Wallis Test Summary | |
|---|---|
| Total N | 100 |
| Test Statistic | 0.714 |
| Degree Of Freedom | 4 |
| Asymptotic Sig. (2-sided test) | 0.950 |

Notation: Multiple comparisons are not performed, because the overall test does not show significant differences across samples.

However, in the periods of identified market declines caused by different crises, different changes in portfolio returns are noticeable. The largest average decline in value of all portfolios was recorded in 2002. Although the minimax portfolio also lost 17% of its value during this period, it is significantly better than other portfolios (value decline from 25% to 28%). With the stabilization of the market in 2003, other portfolios became competitive. During the 2007–2008 crisis, the portfolio created with the minimax model outperformed the other portfolios again. In fact, the return of the minimax portfolio was almost unaffected by the crisis. For only 8 weeks in 2007, the value of the portfolio was slightly lower than at the beginning of the year. All other portfolios failed to return to their initial value after February 2007 until the end of the year.

The same effect can be observed as a consequence of the crisis caused by the COVID-19 pandemic. The differences were more pronounced in the case of the 2007–2008 financial crisis. In late February and beginning of March 2020, the STOXX Europe 600 market index lost more than a third of its value in just four weeks, falling to 67% of its value compared to the beginning of the year (Figure 2). The portfolio values created with the MV model and MAD and Naïve diversification experienced an almost identical decline. On the other hand, the minimax portfolio lost 15% of its value, as with the portfolio created with the CVaR model. Already at the end of May, i.e., 8 weeks after the big market crash, the value of the minimax portfolio exceeded the value at the beginning of the year. The value of the STOXX Europe 600 Index did not reach the value from the beginning of the year until the end of 2020, and the Naïve diversification, MV1, and CVaR portfolio reached the value from the beginning of the year 12 weeks after the minimax portfolio. The MV model with the highest Sharpe ratio reached that value just after 25 weeks. This can be seen in Figure 2, where we present a fixed-based index value of selected portfolios, where the first week for each year is the base period.

We can see that the returns of the selected portfolios are mostly highly positively correlated in the year when the world met financial crisis or the COVID-19 crisis. Figure 2

shows the change in value of the different portfolios during these periods. In 2002, all correlation coefficients between portfolio values were at least 0.92. In 2007 and 2008, the correlation coefficients were at least 0.50, excluding the CVaR portfolio. Except for the portfolio created to minimize the MAD, during the COVID-19 crisis year, no single correlation coefficient between portfolios was less than 0.91, which supports the thesis put forward by Campbell et al. (2002) that, in times of crisis, the correlation coefficients converge toward 1 and the measurement of covariance does not make sense. In all other years, the correlation coefficients were lower, and between some portfolios, even negative, in the years of market growth. Correlation matrices can be found in Appendix B.

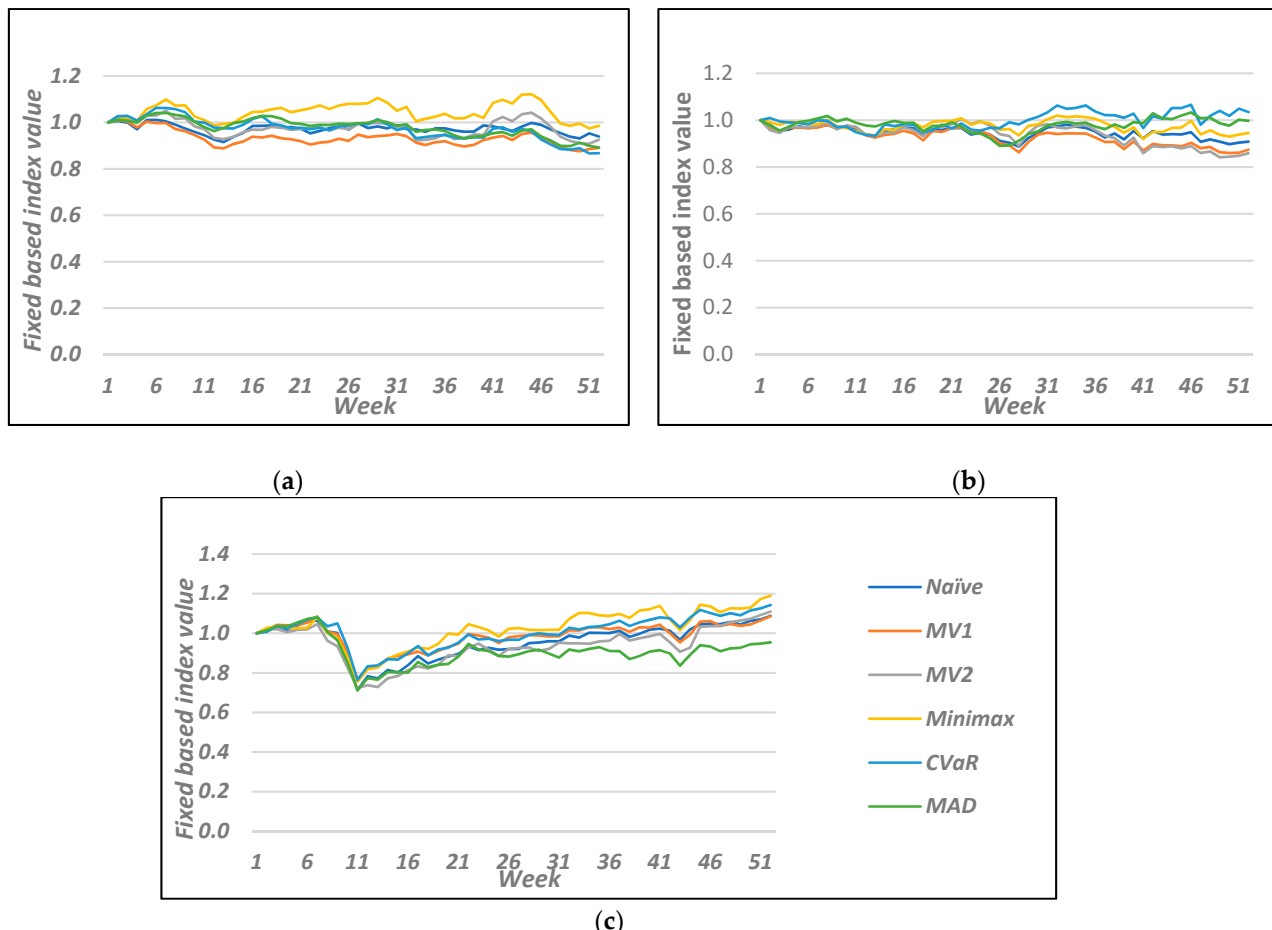

**Figure 2.** Weekly calculated fixed based index for portfolio value by year for a period of market decline: (**a**) 2007, (**b**) 2008, (**c**) 2020.

*4.3. Portfolio Stability Analysis*

We verified the stability of the portfolio by calculating the Sharpe ratio for each portfolio in the individual verification period, year by year. Within each year, we ranked portfolios from 1 to 5 according to the Sharpe ratio, from best to worst (Appendix C). According to this criterion, the portfolio created with the minimax model was best in 7 of 20 periods with an average rank of 2.7. We tested the difference in ranks using the Kruskal–Wallis one-way ANOVA on the ranks test (Table 5). The test rejected the hypothesis that the distribution of ranks was equal. In addition, a pairwise comparison of the model showed statistically significant differences in stability between minimax and MV1, minimax and MV2, CVaR and MV1, MAD and MV1, and MAD and MV2 at all conventional levels of significance. Significance values were adjusted by the Bonferroni correction for multiple tests (Table 6).

**Table 5.** Results of Kruskal–Wallis one-way ANOVA on ranks test for Sharpe ratio ranking.

| Independent-Samples Kruskal–Wallis Test Summary | |
| --- | --- |
| Total N | 100 |
| Test Statistic | 16.308 |
| Degree Of Freedom | 4 |
| Asymptotic Sig. (2-sided test) | 0.003 |

**Table 6.** Results of pairwise comparisons of model for Sharpe ratio ranking.

| Pairwise Comparisons of Model [1] | | | | | |
| --- | --- | --- | --- | --- | --- |
| Sample 1-Sample 2 | Test Statistic | Std. Error | Std. Test Statistic | Sig. | Adj. Sig. |
| MAD-Minimax | 6.100 | 9.149 | 0.667 | 0.505 | 1.000 |
| MAD-CVaR | 9.725 | 9.149 | 1.063 | 0.288 | 1.000 |
| MAD-MV2 | 23.300 | 9.149 | 2.547 | 0.011 | 0.109 |
| MAD-MV1 | 31.875 | 9.149 | 3.484 | 0.000 | 0.005 |
| Minimax-CVaR | −3.625 | 9.149 | −0.396 | 0.692 | 1.000 |
| Minimax-MV2 | 17.200 | 9.149 | 1.880 | 0.060 | 0.601 |
| Minimax-MV1 | 25.775 | 9.149 | 2.817 | 0.005 | 0.048 |
| CVaR-MV2 | 13.575 | 9.149 | 1.484 | 0.138 | 1.000 |
| CVaR-MV1 | 22.150 | 9.149 | 2.421 | 0.015 | 0.155 |
| MV2-MV1 | 8.575 | 9.149 | .937 | 0.349 | 1.000 |

[1] Each row tests the null hypothesis that the Sample 1 and Sample 2 distributions are the same.

*4.4. Portfolio Predictive Power Analysis*

Let us now consider the predictive power of each model. We performed this in two ways. First, for each model, we calculated the average return week by week realized over 52 weeks in the following verification year and compared it to the expected return for the MV model and MAD model, CVaR value for the CVaR model, and the lowest expected return for the minimax model. The realized returns of portfolios created using the MV model converge to an expected value or a value that is close to the expected value 15 to 20 weeks after portfolio creation. In the first few weeks after portfolio creation, returns are extremely volatile. This means that investors need to hold the portfolio for at least 15 to 20 weeks to achieve the expected returns. However, even that does not apply to all portfolios at the efficient frontier. The model based on maximizing the Sharpe ratio proved to be the least accurate of the selected models. In all observed time periods, the deviation from the expected return was significantly negative for this model. A model based on minimization of variance and the MAD model obtained similar results. As deviations of the actual return from the estimated return of the model do not follow the normal distribution, we tested the differences using the Kruskal–Wallis one-way ANOVA on the ranks test for each year. The test rejected the hypothesis that the distribution of differences between expected and actual returns is the same for all portfolios (Table 7). In addition, pairwise comparison confirmed the existence of a statistically significant difference between comparing the minimax portfolio with any other.

**Table 7.** Results of Kruskal–Wallis one-way ANOVA on ranks test for model predictions.

| Independent-Samples Kruskal–Wallis Test Summary | |
| --- | --- |
| Total N | 5200 |
| Test Statistic | 19.255 |
| Degree Of Freedom | 4 |
| Asymptotic Sig. (2-sided test) | 0.000 |

However, the minimax model almost always provided returns that exceed the expected minimum return. During the observed 20 year period, the realized return was below the expected minimum return in only 29 weeks or 2.79% of the time. Moreover, in 25 of the 27 time periods, this negative deviation was negligible, less than 0.005, compared to the MV1 model in 631 weeks (60.67%), MV2 in 953 weeks (91.63%), CVaR at 19 weeks (1.83%), and MAD in 727 weeks (69.90%).

The differences were especially significant during the COVID-19 crisis. The minimax model provided better absorption of the crisis and faster recovery. In 2020, all other models had a negative deviation of the actual return from the expected return in over 90% of the weeks. In just 12 weeks during 2020, the minimax portfolio had a value below expectations according to the model. On the other hand, the MV1 portfolio's return was outside the three-standard-deviation rule during 16 weeks and in MV2 even in 39 weeks. Neither the CVaR nor the MAD model showed better predictive power in the crisis year, with returns 22 and 44 times below the benchmark, respectively. Thus, we can conclude that the minimax goal function value represents a safety measure for the investor, especially at a time when the whole market is declining, and other models show larger deviations from expected performance.

We also calculated how many standard deviations the minimum return obtained by the minimax model is distant from the expected return obtained by the mean-variance model. The minimum return was always in the range of 0.78 to 2.03 standard deviations from the expected one. As the minimum return refers to the left tail of the distribution, we can compare this value with the Value-at-Risk. If the returns are not normally distributed, we cannot use Value-at-Risk. Thus, we recommend using data about the minimum return from minimax to obtain Value-at-risk.

## 5. Conclusions

Although the mean-variance model is the most common framework for portfolio analysis and selection, the inability to satisfy the model assumptions in practice has allowed the creation of new models based on minimum return as a measure of risk. This risk measure is particularly dominant over variance when returns do not follow a normal distribution. We showed that financial time series exhibit many anomalies relative to the normal distribution and that this assumption is rarely satisfied. Under such circumstances, it is appropriate to use the minimax approach, which is an attractive alternative to the conventional mean-variance approach in portfolio analysis. The results of research conducted on the European capital market during the period from 2000 to 2020 showed higher stability and predictive power of the minimax model compared to the mean-variance model. The realized return was lower than the expected minimum return in less than 3% of the cases during the observed period. The minimax model was particularly dominant in a period when the entire market is declining or in a recovery period. These results are in favor of investors who have a pronounced risk aversion and want to reduce the probability of high loss. In addition, based on the results of this study, we can also conclude that the minimax goal function value, or minimum return from the minimax model, can be used as an alternative to the Value-at-Risk method of analysis in further studies when returns are not normally distributed.

**Author Contributions:** Conceptualization, M.S., A.A.-B. and A.Z.; Methodology, A.A.-B.; Software, M.S.; Validation, M.S., A.A.-B. and A.Z.; Formal Analysis, M.S.; Investigation, M.S.; Resources, M.S.; Data Curation, M.S.; Writing—Original Draft Preparation, M.S.; Writing—Review and Editing, M.S., A.A.-B. and A.Z.; Visualization, M.S. and A.A.-B.; Supervision, A.A.-B. and A.Z. All authors have read and agreed to the published version of the manuscript.

**Funding:** This research received no external funding.

**Institutional Review Board Statement:** Not applicable.

**Informed Consent Statement:** Not applicable.

**Data Availability Statement:** The dataset used in this study can be available upon request.

**Conflicts of Interest:** The authors declare no conflict of interest.

## Appendix A

**Table A1.** Results of Kruskal–Wallis one-way ANOVA on ranks test for average number of stocks by portfolio.

| Independent-Samples Kruskal–Wallis Test Summary | |
|---|---|
| Total N | 100 |
| Test Statistic | 17.522 |
| Degree Of Freedom | 4 |
| Asymptotic Sig. (2-sided test) | 0.002 |

**Table A2.** Results of pairwise comparisons of model for Sharpe ratio ranking.

| Pairwise Comparisons of Model [1] | | | | | |
|---|---|---|---|---|---|
| Sample 1-Sample 2 | Test Statistic | Std. Error | Std. Test Statistic | Sig. | Adj.Sig. |
| MAD-Minimax | −5.950 | 9.148 | −0.650 | 0.515 | 1.000 |
| MAD-CVaR | 10.175 | 9.148 | 1.112 | 0.266 | 1.000 |
| MAD-MV2 | 23.100 | 9.148 | 2.525 | 0.012 | 0.116 |
| MAD-MV1 | 33.400 | 9.148 | 3.651 | 0.000 | 0.003 |
| Minimax-CVaR | 4.225 | 9.148 | 0.462 | 0.644 | 1.000 |
| Minimax-MV2 | 17.150 | 9.148 | 1.875 | 0.061 | 0.608 |
| Minimax-MV1 | 27.450 | 9.148 | 3.001 | 0.003 | 0.027 |
| CVaR-MV2 | −12.925 | 9.148 | −1.413 | 0.158 | 1.000 |
| CVaR-MV1 | −23.225 | 9.148 | −2.539 | 0.011 | 0.111 |
| MV2-MV1 | 10.300 | 9.148 | 1.126 | 0.260 | 1.000 |

[1] Each row tests the null hypothesis that the Sample 1 and Sample 2 distributions are the same.

## Appendix B

**Table A3.** Correlation coefficients among portfolio values based on "buy and hold" strategy in 2002.

| 2002 | MV1 | MV2 | Minimax | CVaR | MAD |
|---|---|---|---|---|---|
| MV1 | 1 | | | | |
| MV2 | 0.983195 | 1 | | | |
| Minimax | 0.94487 | 0.945888 | 1 | | |
| CVaR | 0.993267 | 0.975987 | 0.956333 | 1 | |
| MAD | 0.939578 | 0.973847 | 0.930642 | 0.924487 | 1 |

**Table A4.** Correlation coefficients among portfolio values based on "buy and hold" strategy in 2007.

| 2007 | MV1 | MV2 | Minimax | CVaR | MAD |
|---|---|---|---|---|---|
| MV1 | 1 | | | | |
| MV2 | 0.985913 | 1 | | | |
| Minimax | 0.403336 | 0.340911 | 1 | | |
| CVaR | −0.16957 | −0.0977 | 0.271037 | 1 | |
| MAD | 0.597813 | 0.712324 | 0.035441 | 0.365443 | 1 |

**Table A5.** Correlation coefficients among portfolio values based on "buy and hold" strategy in 2008.

| 2008 | MV1 | MV2 | Minimax | CVaR | MAD |
|---|---|---|---|---|---|
| MV1 | 1 | | | | |
| MV2 | 0.838272 | 1 | | | |
| Minimax | 0.410638 | 0.77029 | 1 | | |
| CVaR | 0.782393 | 0.702936 | 0.458958 | 1 | |
| MAD | 0.702328 | 0.586507 | 0.429805 | 0.943927 | 1 |

**Table A6.** Correlation coefficients among portfolio values based on "buy and hold" strategy in 2020.

| 2020 | MV1 | MV2 | Minimax | CVaR | MAD |
|---|---|---|---|---|---|
| MV1 | 1 | | | | |
| MV2 | 0.960801 | 1 | | | |
| Minimax | 0.931854 | 0.907509 | 1 | | |
| CVaR | 0.950296 | 0.938846 | 0.937209 | 1 | |
| MAD | 0.827818 | 0.797511 | 0.590115 | 0.706412 | 1 |

## Appendix C

**Table A7.** Portfolio ranks by Sharpe ratios for each period.

| Year | M1 | M2 | Minimax | CVaR | MAD |
|---|---|---|---|---|---|
| 2001 | 2 | 3 | 1 | 6 | 5 |
| 2002 | 3 | 2 | 4 | 6 | 1 |
| 2003 | 5 | 1 | 3 | 4 | 6 |
| 2004 | 3 | 1 | 2 | 5 | 6 |
| 2005 | 2 | 5 | 1 | 3 | 6 |
| 2006 | 2 | 6 | 1 | 3 | 4 |
| 2007 | 3 | 5 | 1 | 4 | 6 |
| 2008 | 5 | 4 | 1 | 2 | 6 |
| 2009 | 5 | 4 | 3 | 1 | 6 |
| 2010 | 5 | 6 | 1 | 4 | 3 |
| 2011 | 6 | 2 | 4 | 1 | 5 |
| 2012 | 1 | 6 | 4 | 5 | 2 |
| 2013 | 3 | 4 | 2 | 6 | 5 |
| 2014 | 6 | 3 | 5 | 2 | 1 |
| 2015 | 6 | 4 | 2 | 5 | 3 |
| 2016 | 1 | 5 | 4 | 3 | 6 |
| 2017 | 2 | 6 | 5 | 4 | 3 |
| 2018 | 2 | 1 | 6 | 5 | 3 |
| 2019 | 4 | 6 | 5 | 3 | 1 |
| 2020 | 4 | 1 | 2 | 3 | 6 |

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
