# Peer review of "Efficient Asset Allocation: Application of Game Theory-Based Model for Superior Performance"

_ijfs, doi:10.3390/ijfs10010020_

Round 1
Reviewer 1 Report
The manuscript analysis and compares a range of portfolio models based on different risk measures in order to determine potential differences between models and to show that the use of minimax approach is better than mean-variance approach. The study was conducted with data from the last two decades on the European capital market. The results show that in circumstances of crisis there are significant differences and the use of the minimax model to select the portfolios has better predictive efficiency compared to the mean-variance model. The methodology is clearly explained and theoretically grounded and the results give a contribution to the knowledge on the efficient asset allocation.
Author Response
We thank the reviewer for the comments.
Reviewer 2 Report
The study is interesting, actual and based on an appropriate methodology. My comments are intended to help improve the study. I offer the following suggestions for consideration by the authors:
- I suggest reducing the number of keywords - I would remove keywords referring to methodology from the list,
- the introduction should focus only on establishing the topic, so I propose to take out the hypotheses and move them to the methodology section,
- the literature review is rather methodological and does not talk about the effects of crises, especially pandemics, I suggest to expand the literature review in this direction to better ground the topic, as the results and methodology mention the crisis several times,
- it would also be worthwhile to present the software background used in the response in the methodology section,
- I suggest that the methodology be referred to after the tables of the results of the study,
- if possible, the tables in the appendix should be included in the text.
I hope that my suggestions have helped to improve the study.
Author Response
We are grateful to the reviewers for their insightful comments on out paper. We have been able to incorporate changes to reflect most of the suggestions provided by the reviewer. Please see the attachment.
